# Enhanced Expansion of Human Pluripotent Stem Cells and Somatic Cell Reprogramming Using Defined and Xeno-Free Culture Conditions

**DOI:** 10.3390/bioengineering10090999

**Published:** 2023-08-24

**Authors:** Suraj Timilsina, Kaitlyn Faith McCandliss, Evan Trivedi, Luis G. Villa-Diaz

**Affiliations:** 1Department of Biomarkers and Investigative Pathology Unit (BIPU), Charles River Laboratories, Mattawan, MI 49071, USA; timilsinasuraj015@gmail.com; 2Department of Biological Sciences, Oakland University, Rochester, MI 48309, USA; kmccandliss@pm.me; 3Department of Chemistry, Oakland University, Rochester, MI 48309, USA; trivedi@oakland.edu; 4Department of Bioengineering, Oakland University, Rochester, MI 48309, USA

**Keywords:** human pluripotent stem cells, human embryonic stem cells, synthetic polymer grafting, PMEDSAH, self-renewal, reprogramming

## Abstract

Human embryonic stem cells and induced pluripotent stem cells (hPSC) have an unprecedented opportunity to revolutionize the fields of developmental biology as well as tissue engineering and regenerative medicine. However, their applications have been significantly limited by the lack of chemically defined and xeno-free culture conditions. The demand for the high-quality and scaled-up production of cells for use in both research and clinical studies underscores the need to develop tools that will simplify the in vitro culture process while reducing the variables. Here, we describe a systematic study to identify the optimal conditions for the initial cell attachment of hPSC to tissue culture dishes grafted with polymers of N-(3-Sulfopropyl)-N-Methacryloxyethyl-N, N-Dimethylammoniun Betaine (PMEDSAH) in combination with chemically defined and xeno-free culture media. After testing multiple supplements and chemicals, we identified that pre-conditioning of PMEDSAH grafted plates with 10% human serum (HS) supported the initial cell attachment, which allowed for the long-term culture and maintenance of hPSC compared to cells cultured on Matrigel-coated plates. Using this culture condition, a 2.1-fold increase in the expansion of hPSC was observed without chromosomal abnormalities. Furthermore, this culture condition supported a higher reprogramming efficiency (0.37% vs. 0.22%; *p* < 0.0068) of somatic cells into induced pluripotent stem cells compared to the non-defined culture conditions. This defined and xeno-free hPSC culture condition may be used in obtaining the large populations of hPSC and patient-derived iPSC required for many applications in regenerative and translational medicine.

## 1. Introduction

Human pluripotent stem cells (hPSC): human embryonic stem cells (hESC) and human induced pluripotent stem cells (hiPSC), have the unique abilities to replenish themselves through self-renewal and to differentiate into different types of specialized cells [1,2,3]. Since the first report of the derivation of hESCs in 1998 [2] and the successful development of hiPSCs from somatic cells in 2007 [1,3], stem cell research is rapidly progressing from basic research to the clinical arena with the hope that derivatives of these cells can be used to treat and manage debilitating diseases [4,5]. These cells also play essential roles in the study of organogenesis and tissue regeneration and represent an important cell source for cell-based therapies and drug development. Continued growth in cell therapies has created a consensus for the need for specialized and defined materials to translate stem cell studies into clinical trials.

Many of the current culture conditions are still based on undefined support systems that contain xenogeneic components, limiting the ability to use hPSC-derived cells to treat debilitating diseases [6,7,8]. In addition to the risk of transmitting pathogens and the introduction of tumor-derived growth factors, culture conditions that depend on undefined support systems affect experimental reproducibility and the ability to interpret mechanistic studies due to lot-to-lot variations and undefined conditions. This ultimately hinders the transition of hPSC-derived products into a clinical setting.

To overcome these problems, a significant effort has been made to understand the signaling pathways and molecules required to maintain the self-renewal and pluripotent state of hPSCs, resulting in the formulation of defined and xeno-free culture media. However, as anchorage-dependent cells, hPSCs also require specific extracellular matrices (ECM) as substrates to maintain their self-renewal [8]. Hence, the optimization of culture systems for hPSCs is not limited to defining the culture medium components, but also includes the substrates and environmental cues, among many other factors. To address these issues, natural or recombinant ECM proteins have been used to maintain hPSC self-renewal [9,10,11]. However, not all ECM proteins are suitable for hPSC culture as they cannot maintain an undifferentiated hPSC population or allow the clonal survival of hPSCs without the use of apoptosis inhibitors. The high production costs, labor-intensive cell passaging, and the limited scale-up potential associated with many of these ECM substrates have driven researchers to seek alternative substrates in the synthetic polymer arena composed of RGD (Arg-Gly-Asp) peptides covalently immobilized on an acrylate coating [12,13,14,15,16]. We reported the first fully defined synthetic polymer coating, which maintained the long-term growth, self-renewal, and pluripotency of hPSCs in a human cell conditioned medium (CM) and a defined serum-free medium, using the grafting of poly [2-(methacryloyloxy)ethyl dimethyl-(3-sulfopropyl)ammonium hydroxide] (PMEDSAH) on tissue culture polystyrene (TCP) dishes [17]. Here, we report an optimization in the protocol through the use of PMEDSAH-grafted (PMEDSAH-g) TCP dishes on the culture and derivation of hPSC, which is compatible with multiple chemically defined and xeno-free culture medium formulations that exist in the market. We also show the successful derivation and long-term culture of hiPSC under defined and xeno-free conditions.

## 2. Materials and Methods

### 2.1. Chemicals and Materials

The following chemicals and materials were used: Angiopoietin-1 (ang1)-based peptide QHREDGS, retinoic acid (RA) (ATRA), integrin-activating antibody, calcium chloride dehydrates, magnesium chloride hexahydrate, manganese (II) chloride tetrahydrate, phorbol 12-myristate 13-acetate (PMA), and human serum (HS) were purchased from Sigma-Aldrich (Sigma-Aldrich Corp., Milwaukee, WI, USA). Fetal bovine serum (FBS), Teflon-coated magnetic 1″ stir bar, and stir plate were purchased from VWR (VWR International, Radnor, PA, USA). Knockout serum replacement (KOSR), anhydrous ethanol, and sodium chloride were obtained from Invitrogen (Thermo Fisher Scientific, Waltham, MA, USA). The monomer N-(3-Sulfopropyl)-N-Methacryloxyethyl-N, N-Dimethylammoniun Betaine was purchased from Monomer-Polymer and Dajac labs (Monomer-Polymer and Dajac labs, Ambler, PA, USA). The TCPS of 35 mm in diameter were purchased from BD Falcon (Thomas Scientific, Swedesboro, NJ, USA). The UV Ozone cleaner (model no. 342) was purchased from Jelight Company (Jelight Company Inc., Irvine, CA, USA), and the transonic 52 ultrasonic cleaner (50/60 Hz) was obtained from Emerson (Emerson Electric, St. Louis, MO, USA). The cylindrical reaction vessel (500 mL), reaction vessel lid (3 × 24/40 necks), Viton O-ring, reactor clamp, support clamp for reaction vessel, vacuum/air manifold, condenser (24/40 Joint, 300-mm Jacket Length), heating mantle, heating mantle controller, and 24/40 glass stopper were purchased from Chemglass (Chemglass Life Science, Vineland, NJ, USA). The following xeno-free and chemically defined media were used for the hPSC culture: StemFlex (Gibco^TM^ ThermoFisher Scientific, Waltham, MA, USA), mTeSR^TM^ Plus (Stemcell^TM^ Technologies, Vancouver, Canada), PluriSTEM^TM^ (Sigma-Aldrich Corp.), and StemFit (Reprocell, Beltsville, MD, USA) (Appendix A).

### 2.2. Synthetic Surface Preparation on TCP Dishes of PMEDSAH-g Dishes

The UV ozone-initiated free radical polymerization was carried out in a fume hood with connections for nitrogen gas. Each reaction was used to prepare 24 TCP dishes, and this procedure was reproduced 10 times to produce 240 PMEDSAH-grafted plates. The monomer solution consisting of 0.25 M MEDSAH was dissolved in a mixture of deionized water and ethanol (4:1, *v*/*v*) in a 500 mL reaction vessel. The solution was degassed for 60 min through nitrogen purge. Then, the monomer solution was heated to 72 °C. While the reaction vessel was being heated, TCP dishes were activated through UV ozone treatment for 45 min to create initiation sites on the surface. After activation, the dishes were transferred to the reaction vessel and the temperature was raised to 76–80 °C (Appendix A). The surface-initiated polymerization occurred over a 2 h time period under a nitrogen atmosphere at 76–80 °C. Once the process was completed, grafted dishes were removed from the reaction vessel and rinsed with 1% saline (*v*/*v*) solution at 50 °C, followed by ultra-sonication in DI-water and air drying inside the hood.

### 2.3. Preparation and Use of 10% Human Serum (HS) Treated (HSt)-PMEDSAH-g Dishes

Freshly prepared PMEDSAH-g dishes were treated with 10% HS (*v*/*v* DMEM/F12) for 30 min at room temperature, followed by a couple of washes with cold sterile Dulbecco’s phosphate buffered saline (D-PBS). Two sets of 10% HSt-PMEDSAH-g dishes were prepared. The first set, labeled as wet (W10% HSt-PMEDSAH-g), was used right after its preparation for culturing hPSC for 20 consecutive passages. For the other set of 10% HSt-PMEDSAH-g dishes, after their washes with cold sterile D-PBS, the dishes were air dried and wrapped with parafilm and stored at 4 °C until their use for cell culture, and were labeled as dry (D10% HSt-PMEDSAH-g). These plates were used for weekly passaging of hPSC for 13 consecutive passages. In both conditions, after every fifth passage, the cells were characterized for self-renewal and pluripotency.

### 2.4. Preparation of Matrigel-Coated Dishes

Matrigel hESC-qualified (Corning, Glendale, AZ, USA) was diluted to a concentration of 0.1 mg/mL in cold Dulbecco’s modified Eagle’s medium/F12 (DMEM/F12; GIBCO), which was then applied to TCP dishes. This coating was allowed to polymerize at room temperature during the 2 h incubation. Excess Matrigel-DMEM/F12 solution was aspirated, and the dishes were washed with cold sterile D-PBS followed by seeding of the cells.

### 2.5. Human Pluripotent Stem Cells (hPSC) Culture

NIH-approved hESC lines H1 and H9 (WiCell Research Institute, Madison, WI, http://www.wicell.org (accessed on 31 October 2022), and three hiPSC derived in our laboratory [18] (hGF2-iPSCs, hGF4-iPSCs, hFF [human foreskin fibroblasts] iPSC) were cultured on 10% HSt-PMEDSAH and Matrigel-coated dishes at 37 °C with 5% CO_2_ using chemically defined media (Appendix A). Colony growth and development were observed every 48 h and the differentiated cells were mechanically removed using a sterile pulled-glass pipet under a stereomicroscope (LeicaMZ9.5, Leica Microsystems Inc., Buffalo Grove, IL, USA). Undifferentiated colonies were passaged weekly using L7-hPSC passaging solution (Lonza) and with supplementation of 10 μm of ROCK inhibitor (Sigma) [19] at the freshly passaged cells. After dissociation into single cells, cell counting was performed, and 10,000 cells (1000 cells/cm^2^) were plated on 10% HSt-PMEDSAH-g dishes and Matrigel-coated dishes and cultured for 7 days. The chemically defined culture medium was replaced every other day. Transmitted light images of cells and colonies were taken using an EVOS^®^FL Cell Imaging System (Thermo Fisher Scientific).

### 2.6. Quantitative Analysis of the Total Cell Number

During 5 initial and consecutive weekly passages, the total number of cells grown on each plate of 10% HSt-PMEDSAH and Matrigel was counted and recorded at the time of passage for comparative analysis. From the total number of cells obtained, 10,000 cells were used per passage into a new substrate. A theoretical yield of the total cell number of hPSC obtained on different substrates was calculated assuming that all cells would be passaged each week instead of the only 10,000 single cells that were seeded. The theoretical yield of cells was determined with the formula CN_(*n*+1)_ = CN_*n*_ × TN_(*n*+1)_/10,000, in which CN is the calculated total cell number, TN is the total cell number and *n* is the passage number.

### 2.7. Immunofluorescence Staining

Cells grown on 10% HSt-PMEDSAH-g dishes were fixed in 4% paraformaldehyde for 10 min at room temperature followed by permeabilized with 0.1% Triton X-100 for 10 min. Primary antibodies raised against OCT3/4 (Cell signaling), SOX2 (MilliporeSigma, Burlin-tong, MA, USA), NANOG (Abcam, Cambridge, UK), TRA-1-60 (Abcam), and TRA-1-81 (MilliporeSigma) were diluted in 1% normal donkey serum and incubated overnight at 4 °C with gentle shaking and detected with respective secondary antibodies. Micrographs were captured using an EVOS®FL Cell Imaging System (Thermo Fisher Scientific).

### 2.8. Flow Cytometry Analysis

Cells were washed with D-PBS and harvested through incubation in L7-hPSC passaging solution. After this, cells were incubated for 30 min in dark at 4 °C, first with human IgG to block un-specific binding and then with human/mouse NANOG APC-conjugated antibody (Biolegend, San Diego, CA, USA), OCT4 PE-conjugated antibody (Biolegend), SSEA3 APC-conjugated antibody (R&D systems), SSEA4 PE-conjugated antibody (R&D systems, Minneapolis, MN, USA), TRA-1-60 FITC-conjugated antibody (Biolegend), and TRA-1-81 PE-conjugated antibody (Biolegend). At least 10,000 events were acquired for each sample using the BD FACSCanto II and BD FACSAria (BD Biosciences) instruments and the flow cytometry data were analyzed using the FlowJo software (https://www.flowjo.com/, accessed on 31 October 2022).

### 2.9. RNA Isolation and Quantitative Real-Time PCR

RNA extraction from cells was conducted by directly adding Trizol (Thermo Scientific) into the culture dishes. RNA was isolated and purified using the Direct-zol™ RNA Miniprep (Zymo Research, Irvine, CA, USA) following the manufacturer’s protocol. RNA quality and concentration were obtained using a NanoDrop™ 2000c Spectrophotometer (Thermo Scientific). Reverse transcription from 1 µg of total RNA into cDNA was performed using SuperScript™ VILO™ cDNA Synthesis Kit (Invitrogen™, Waltham, MA, USA). Quantitative PCR was then performed using TaqMan probes (Applied Biosystems, Waltham, MA, USA) listed in Appendix A and TaqMan Universal PCR Master Mix (Applied Biosystems) on a 7900 HT Fast Real-Time PCR system (Applied Biosystems). Gene expression data were normalized to the expression levels of GAPDH, and the relative gene expression values were calculated using the delta-delta *cT* method.

### 2.10. Embryoid Body (EB) Formation and Analysis of hPSC Self-Renewal and Pluripotency

Formation of EB from hPSC passage 20 cultured on W10% HSt-PMEDSAH-g dishes was conducted using cell clusters cultured in suspension in 10% FBS/DMEM alpha (Life Technologies, Carlsbad, CA, USA) on ultra-low cell attachment dishes (Corning, Glendale, AZ, USA) for 10 days to promote differentiation. Quantitative analysis of self-renewal and pluripotency assessed by the trilineage differentiation potential of hPSC cultured on W10% HSt-PMEDSAH-g dishes at passage 20 was conducted using TaqMan hPSC Scorecard Assay (Thermo Scientific) following the manufacturer’s protocol. Briefly, total RNA from undifferentiated hPSC cultured on W10% HSt-PMEDSAH-g dishes and EB obtained from the same pool of hPSC was extracted and used to generate the respective cDNA, as described above. The respective cDNA samples, TaqMan Fast Advanced Master Mix Template, and RNase-free water were combined and reconstituted in each well of two 96-well dishes that contains 94 predefined TaqMan Gene Expression assays (including endogenous controls) dried-down in the wells TaqMan^®^ hPSC Scorecard™ Panel by adding 10 µL of the reaction mixture per well. Each plate was loaded and run using the StepOnePlus Real-Time PCR System (Thermo Fisher Scientific) with the fast thermal cycling condition. The list of genes analyzed is presented in Appendix A. The gene expression data were analyzed using the web-based hPSC Scorecard™ Analysis Software to confirm the pluripotency of the samples and predict their differentiation potential and outcome by comparing them to a common reference set.

### 2.11. Cytogenetic Evaluation

After 20 weeks of cell culture, standard G-band analysis (Karyotype analysis) was performed on cells cultured on W10% HSt-PMEDSAH-g dishes by a cytogenetic specialist at the WiCell Institute (Madison, WI, USA). Chromosomes were prepared using the standard protocols and measurements were performed using the Giemsa/Trypsin/Leishman (GTL)-banding method on at least 20 metaphase preparations.

### 2.12. Reprogramming of Human Somatic Cells into iPSC on 10% HSt-PMEDSAH-g Dishes and Quantification of Reprogramming Efficiency

For iPSC reprogramming, 3  ×  10^5^ human gingival fibroblasts (hGF) were seeded on W10% HSt-PMEDSAH-g dishes in high-glucose DMEM supplemented with 10% HS (Sigma), 1% nonessential amino acids (NEAA) (Gibco Thermo Fisher Scientific), 1% GlutaMax (Gibco Life Technologies), and 1% penicillin-streptomycin (Gibco Life Technologies). When cell confluence reached about 50%, the Sendai viral vectors carrying KLF4, SOX2, OCT3/4, and c-MYC (CytoTune kit 2.0, Thermo Fisher Scientific) supplemented with 10 µg of hexadimethrine bromide (Polybrene; Sigma-Aldrich) were used for reprogramming into iPSC using the standard protocols [18]. Eight hours later, fresh medium was added. Seventy-two hours later, the cells were sub-cultured into a set of six dishes each for W10% HSt-PMEDSAH-g and Matrigel-coated, and then twenty-four hours later, the medium was replaced with StemFlex medium (Thermo Fisher Scientific) supplemented with 10 µM Y-27632 dihydrochloride (ROCK inhibitor) (Stem Cell Technologies, Vancouver, CA, USA). The hPSC culture medium with ROCK inhibitor was refreshed every day until week 3 or 4 until iPSC colonies appeared. The human iPSC colonies were analyzed by the alkaline phosphatase (AP) activity, and together with the colony morphology (well-defined borders and high nuclei per cytoplasmic ratio), were used to identify reprogrammed hiPSC colonies. The number of hiPSC colonies obtained on the W10% HSt-PMEDSAH-g dishes and Matrigel-coated dishes were quantified and used to calculate the respective reprogramming efficiency as follows: the number of hiPSC colonies in each group was divided by 3  ×  10^5^, which was the original number of fibroblasts plated for each group before reprogramming. In addition, the hiPSC were further characterized for self-renewal and pluripotency potential through immunostaining, flow cytometry, and qRT-PCR analysis.

### 2.13. Alkaline Phosphatase (AP) Staining

Undifferentiated colonies were identified through specific AP staining, which was conducted using the AP Staining Kit (Reprocell). The cells were fixed with fixing solution for 2–5 min, then rinsed and incubated in AP Substrate Solution in the dark at room temperature for 15 min. The cells were rinsed and covered with 1 X D-PBS to prevent drying before quantitative analysis. 

### 2.14. Data Analysis

Three independent replicates for each experiment were performed and data were expressed as mean value ± SD. Two data sets were compared using the unpaired student *t*-test function to calculate *p* values. The levels of statistical significance were set at *p* < 0.05 (* = *p* < 0.05, ** = *p* < 0.005, *** = *p* < 0.0005).

## 3. Results

### 3.1. Optimization and Standardization of PMEDSAH Grafting (PMEDSAH-g) and hPSC Culture

We synthesized PMEDSAH-g TCP dishes using a surface-initiated graft polymerization procedure through optimization into a one-day protocol, as described in detail in the methods section (Figure 1 and Appendix A). To standardize the hPSC culture conditions on the chemically inert synthetic polymer-coated TCP dishes using defined and xeno-free conditions, we performed a systematic study to optimize the initial cell attachment. Our systematic study and analysis included the activation of integrins using the peptide (QHREDGS) [20,21] or the integrin-activating antibody clone HUTS-4 [22,23] by altering the pH of the media, as suggested and supported by previous publications [24,25], supplementing divalent cations and certain cytokines to the media to activate cell-ECM adhesion signaling [26,27,28,29,30], in addition to treating (pre-conditioning) the freshly prepared PMEDSAH-g dishes with graded concentrations (1%, 10% and 100%) of fetal bovine serum (FBS) [31], Knockout™ Serum Replacement (KOSR) [7], and human serum (HS) [32,33]. For all these conditions, the initial cell attachment, the development of undifferentiated colonies, and the number of consecutive passages that were sustained were analyzed. The results are summarized in Table 1. Among all the treatments on the PMEDSAH-g dishes and the supplements in the chemically defined media (Appendix A), the PMEDSAH-g dishes were preconditioned with 10% human serum (W10% HSt-PMEDSAH-g) in DMEM/F12 for 30 min at room temperature, followed by a couple of washes with cold D-PBS. These processes produced the following results in the hPSC culture: improved cell attachment and long-term culture (20 consecutive passages) in an undifferentiated state (Figure 2). The hPSC cultured in this condition retained its pluripotency, as demonstrated by the ability to differentiate into endoderm, ectoderm, and mesoderm derivatives, and maintained a normal karyotype (Figure 3). On the other hand, we were not able to obtain a stable attachment and growth of the hPSC on the PMEDSAH-g dishes using media supplemented with other graded concentrations of HS, and the treatment of TCP dishes with 10%-HS did not support the attachment and growth of hPSC, as observed in the 10% HSt-PMEDSAH-g dishes (Appendix A).

### 3.2. Culture and Expansion of hPSC on 10% HSt-PMEDSAH-g Dishes

We cultured hPSC on freshly prepared W10% HSt-PMEDSAH-g dishes, and tested these dishes until passage 20, and on D10% HSt-PMEDSAH-g dishes, which were prepared in lots and stored at 4 °C for weakly passaging and tested until passage 13 (Appendix A). Undifferentiated hPSC colonies were identified by observing the colonies with a compact morphology and defined border and by using alkaline phosphatase (AP) staining [34]. Interestingly, in the initial experiments, the number of undifferentiated colonies and the total cell number (ratio) were higher on W10% HSt-PMEDSAH-g when compared to the Matrigel control group (Figure 2); however, it was not statistically significant. The attachment and growth of hPSC were continuously observed to be better on the W10% HSt-PMEDSAH-g dishes compared to the control group (Matrigel), as shown in the micrographs in Figure 2. To quantify the impact of the attachment and long-term expansion of hPSCs on the W10% HSt-PMEDSAH-g dishes, we calculated the theoretical yield of the total hPSC potential obtained after five passages on the Matrigel and W10% HSt-PMEDSAH-g dishes, assuming that all cells at each passage were sub-cultured, instead of the 10,000 cells that were propagated at each passage. It was estimated that on the W10% HSt-PMEDSAH-g dishes, the expansion of 10,000 cells over a period of five weeks would yield up to 4.9 × 10^8^ undifferentiated stem cells. This level of cellular expansion on the W10% HSt-PMEDSAH-g dishes was 2.1-fold greater (*p* < 0.0001) than the theoretical yield calculated when cultured on the Matrigel-coated dishes, as indicated in the graph in Figure 2**.** The total cell numbers observed demonstrated that W10% HSt-PMEDSAH-g facilitated the attachment and long-term growth and expansion of hPSC to a significant extent.

### 3.3. Characterization of hPSC Cultured on W10% HSt-PMEDSAH-g Dishes

The hPSC cultured on W10% HSt-PMEDSAH-g dishes were characterized for self-renewal, pluripotency, and genomic stability. The colonies of hPSC cultured on the W10% HSt-PMEDSAH-g dishes were observed to be strongly positive for undifferentiated hPSC markers including NANOG, OCT4, SOX2, TRA 1-60, and TRA 1-81, as shown by the immunofluorescence staining micrographs at passages 5, 10, 15, and 20 (Figure 3). About 99% of the cells show positive co-expression of SSEA3 and SSEA4 with OCT4, as revealed by the flow cytometry dot plots (Figure 3B). Similarly, the hPSC cultured on the D10% HSt-PMEDSAH-g dishes and stored at 4 °C were also strongly positive for NANOG, OCT4, and SOX2, as shown by the immunofluorescence staining (Appendix A), and strongly co-expressed TRA-1-60 and TRA-1-81 with NANOG, as observed in the flow cytometry dot plots (Appendix A). All of these data were obtained using the chemically defined and xeno-free medium StemFlex. Comparable results were obtained using StemFit, mTeSR Plus, and PluriSTEM-XF—all for chemically defined and xeno-free culture media—as shown by the immunostaining for NANOG, OCT4, SOX2, and SSEA4 (Appendix A).

The pluripotency of the hPSC cultured on the W10% HSt-PMEDSAH-g dishes was determined using an embryoid body (EB) formation assay after 20 passages. Analysis of the EBs indicated the expression of representative genes of the endoderm, mesoderm, and ectoderm lineages. Quantitative analysis of the self-renewal and pluripotency was assessed by the trilineage differentiation potential of the hPSC cultured on the W10% HSt-PMEDSAH-g dishes using a TaqMan hPSC Scorecard Assay based on real-time qPCR assays (Figure 3). All of these findings demonstrated that W10% HSt-PMEDSAH-g supported the self-renewal and pluripotency of hPSC. We quantitatively validated the confirmed pluripotency (trilineage differentiation potential) of the cells cultured on the W10% HSt-PMEDSAH-g dishes using real-time qRT-PCR assays for several genes representing three germ lineages (Appendix A).

The genetic stability of the cells cultured on the W10% HSt-PMEDSAH-g dishes was evaluated through standard G-band analysis after 20 passages as chromosomal changes might occur during the long-term culture and maintenance of hPSC [8,35,36,37]. Our results confirmed a normal human karyotype (Figure 3).

### 3.4. Reprogramming of Human Somatic Cells on W10% HSt-PMEDSAH-g Dishes

Reprogramming of patient-derived cells in chemically defined and xeno-free conditions will enhance the successful use of these cells in translational medicine to treat patients’ debilitating diseases. Thus, we investigated whether W10% HSt-PMEDSAH-g could have a positive impact on the reprograming of fibroblasts into hiPSC (Figure 4). Sendai viral vectors carrying the *KMOS* cocktail were used to infect human gingival fibroblasts in both controls (Matrigel-coated dishes) and experimental groups (10% HSt-PMEDSAH-g dishes). On day 28, the hiPSC colonies were classified as successfully reprogrammed or as pre-iPSCs using the following criteria: reprogrammed colonies should have positive AP staining, well-defined borders, and cells should have a high nucleus: cytoplasm ratio. We characterized the hiPSC developed on the 10% HSt-PMEDSAH-g dishes as self-renewal and pluripotent. The hiPSC were strongly positive for NANOG, OCT4, and SOX2, as revealed by our immunostaining, and ~99% positive for SSEA3, SSEA4, TRA 1-60, and TRA 1-81, as shown by the flow cytometry histograms (Figure 4). Similarly, the pluripotency of the hiPSC obtained on the W10% HSt-PMEDSAH-g dishes was determined by EB formation and we confirmed their tri-lineage differentiation potential using real-time qPCR assays for several genes representing all germ lineages (Figure 4). As expected, the hiPSC developed in the control group (Matrigel) were also self-renewing and pluripotent (Figure 5 and Appendix A).

We quantified the hiPSC colonies developed on the Matrigel and W10% HSt-PMEDSAH-g dishes through AP staining and manually counting the colonies (Figure 5). Interestingly, the reprogramming efficiency was significantly higher (*p* < 0.0068) in the W10% HSt-PMEDSAH-g dishes compared to the control group—0.37% vs. 0.22%, respectively (Figure 5).

## 4. Discussion

Owing to the intrinsic capability for unlimited self-renewal and the ability to make all the cells in the body, pluripotent stem cells (PSC) are an ideal candidate to be used as a starting material for cell therapies. Much effort has been put into creating methods for human PSC (hPSC) expansion that are defined, robust, simple, and safe. The evolution of hPSC culture from feeder-cell dependence and non-defined conditions to feeder-free and defined microenvironments has been enabled by the development of new culture materials [8]. Thus, the development of a standard hPSC culture method using a defined and xeno-free culture environment that can support hPSC self-renewal, pluripotency, and long-term propagation will advance our knowledge of PSC biology and increase the effectiveness of hPSC expansion under defined conditions for potential human applications. In this study, we established a chemically defined and xeno-free culture system for the long-term culture and maintenance of hPSC while preserving the key molecular and functional features of hPSC. Furthermore, the culture system described here also permitted the derivation of xeno-free and transgene-free hiPSC. In this regard, the culture of hPSC in chemically defined and xeno-free conditions will contribute substantially to future biotechnological and medical applications.

Our original protocol was based on culturing hPSC on PMEDSAH-grafted (PMEDSAH-g) TCP dishes using a human cell-conditioned commercial medium [17]. Herein, we report a chemically defined and xeno-free culture of hPSC on PMEDSAH-g dishes pre-conditioned with 10% human serum (10% HSt-PMEDSAH-g). We optimized the PMEDSAH grafting of the TCP dishes and standardized their use for the cultivation of hPSC. Our standardization of the cultivation of hPSC on PMEDSAH-g dishes primarily focused on finding a reliable method to stimulate the initial adherence of hPSC and their long-term maintenance in xeno-free conditions. For this, we performed a systematic study through supplementing and pre-conditioning PMEDSAH-g dishes. We found that pre-conditioning the PMEDSAH-g dishes with 10% HS in DMEM/F12 gave consistent results for the attachment, long-term culture, and maintenance of hPSC.

To maintain hPSC pluripotency, a stem cell niche requires not only growth factor signals, but also cell adhesion. Without the support of exogenous adhesion proteins, hPSC either die or differentiate [7,38]. Hence, hPSC are anchorage-dependent cells utilizing integrins for adhesion to substrates, and it is also known that integrin signaling is important in hPSC cell adhesion, self-renewal, and the maintenance of pluripotency [39,40,41,42,43,44]. Thus, we employed several methods to activate integrin signaling and tested the effect on the cell adhesion and maintenance. It has been suggested and reported that angiopoietin-1 can directly support cell adhesion mediated by integrins [20,21,45]. Also, activation antibodies are thought to facilitate cell adhesion by converting low-affinity integrin molecules to a high-affinity state through direct binding and the induction of conformational changes [22,23,45]. In addition, several publications have also reported that the attachment of mammalian cells to a solid substrate is followed by an increase in the intracellular pH, which is mediated by the Na+/H+ antiporter, and this is obligatory for the development of cell adhesion [24,25,46]. Based on these findings, we tested the peptide motif (QHREDGS) derived from angiopoietin-1, the integrin binding site in angiopoietin-1, to promote cell adhesion [20,21,47]. We also tested the compatibility of integrin-activating antibodies and tested the commercial media used in this project with varying pH with the aim to promote and stabilize cellular adhesion on the PMEDSAH-g dishes. However, we were not able to observe any improvement in the cell attachment using these supplements.

Furthermore, several reports indicate that divalent cations, including Ca^2+^, Mg^2+^, and Mn^2+^, control cell-substrate adhesion and laminin expression [26,27,48,49]. However, the supplementation of additional divalent cations into the media used for the culture of hPSC on the PMEDSAH-g dishes did not support initial cell adhesion. Others have reported that cytokines regulate cell proliferation and cell adhesion to cell substrates [29,46]. We tested retinoic acid (RA) and phorbol myristate acetate (PMA), which were previously reported to enhance mammalian cells, including hPSC, adhesion to matrix [30,50,51]. Like other supplements tested, our results indicated that cytokine supplementation did not support the cell attachment and proliferation of undifferentiated hPSC into the PMEDSAH-g dishes using a chemically defined and xeno-free media.

To keep cells alive for longer periods of time and to evaluate proliferation, migration, and differentiation, basal medium need to be supplemented with several factors. Among several such factors, serum derived from animals or humans is commonly used to maintain and proliferate cells. Fetal bovine serum (FBS), used in cell expansion protocols, provides vital nutrients, attachment factors, toxin scavengers, and growth factors, which are essential for the growth and maintenance of cells. We observed that the supplementation of FBS to the hPSC culture media promoted the attachment of cells on the PMEDSAH-g dishes; however, it also induced cell differentiation. In addition, it is also unsuitable for therapeutic purposes due to its ill-defined nature, high lot-to-lot variable composition, the risk of transmitting infectious agents, and its immunizing effects [31]. Therefore, a continuous search for alternative sources for human cell cultivation that replaces the FBS is ongoing. Knockout™ Serum Replacement (KOSR), on the other hand, is used for the serum-free culture of PSC from multiple species. This defined serum-free formulation directly replaces FBS in the existing protocols and alleviates many of the drawbacks of using FBS in stem cell culture, including the differentiation of cells and heat-inactivation [7]. However, during our investigation, KOSR also failed in promoting the cellular adhesion and growth of hPSC on the PMEDSAH-g dishes. Recent studies have suggested that human body fluids, including autologous serum, allogeneic human serum (HS), umbilical cord blood serum, platelet derivatives, and follicular fluid, might be promising substitutes [32,52]. However, there have been controversial issues concerning the advantages of HS, and detailed studies about the effects of HS on hPSC have not been conducted. Through our systematic study, with a gradient concentration of HS, we reported that the supplementation of hPSC culture media with 10% of HS improved the cell adhesion, self-renewal, and growth compared to FBS. Interestingly, the pre-conditioning of the freshly prepared PMEDSAH-g dishes with 10% HS in DMEM/F12 (10% HSt-PMEDSAH-g) under both the conditions—W10% HSt-PMEDSAH-g and D10% HSt-PMEDSAH-g—supported better attachment, growth, and long-term maintenance of hPSC. Furthermore, we successfully used several chemically defined and xeno-free commercial media to cultivate hPSC on 10% HSt-PMEDSAH-g dishes. We recognized that HS is susceptible to batch-to-batch inconsistency; however, during these studies, several batches of HS were tested and all of them supported the initial cell attachment of the hPSCs to the PMEDSAH-g dishes with fully defined chemical media. Further efforts on our side will include identifying the key components present in the HS supporting the initial attachment of hPSCs to PMEDSAH-g dishes.

As Matrigel is the most commonly used feeder-free substrate for hPSC culture, it was used as a control in our experiments. A key concern in hPSC research is maintaining the pluripotent cultures in an undifferentiated and proliferative condition without any potential chromosomal aberrations. We found that the W10% HSt-PMEDSAH-g led to a higher number of undifferentiated colonies and total number of cells compared to the experimental group. In addition, the theoretical yield of the total number of undifferentiated hPSC was observed to be significantly higher under the W10% HSt-PMEDSAH-g culture condition compared to the control group. The hPSC cultured on the freshly prepared W10% HSt-PMEDSAH-g dishes and D10% HSt-PMEDSAH-g dishes retained their self-renewal, pluripotency, and a normal karyotype after multiple passages during culture, which indicates that the hPSC maintained their unique characteristics and genomic stability. We showed this response of the culture of hPSC on the 10% HSt-PMEDSAH-g dishes using multiple chemically defined commercial media (Stemflex, StemFit, mTSeR Plus, and PluriSTEM-XF). These findings are well aligned with the long-term goal of the large-scale production of clinical-grade hPSC and hiPSC. In addition to significantly higher efficiency in hPSC expansion during long-term culture, the 10% HSt-PMEDSAH-g dishes have the following advantages compared to Matrigel: defined composition, long-term storage stability with negligible lot-to-lot variability, easy preparation, and compatibility with standard sterilization techniques. All of these features make the 10% HSt-PMEDSAH-g dishes a very promising substrate to obtain scalable populations of clinical-grade hPSC. This can be easily accomplished with 10% HSt-PMEDSAH-g dishes while reducing the cell expansion time and production cost, as well as possible contamination and population drift under the current culture conditions.

Induced pluripotent stem cells (iPSC) are obtained after the genetic reprogramming of somatic cells (e.g., fibroblasts and blood cells) into a cell that acquires the characteristics of PSC: self-renewal and the capacity for differentiation in all cell types of the body [53,54,55]. Human iPSC presents a uniquely scalable platform for the study of inherited diseases, cell modeling, and as a novel means for cell replacement in clinical settings, thereby replacing the controversial use of hESC. Despite having tremendous therapeutic potential, the translational research of hiPSC replacement therapy for human patients is relatively slow, highlighting the urgency for improved hiPSC-based cell therapy. Following the pioneering study by the Yamanaka group, others have used different strategies to produce hiPSC to meet the safety and quality criteria for effective therapeutic applications, in addition to increasing the reprogramming efficiency. As we observed significantly higher efficiency in hPSC expansion on the W10% HSt-PMEDSAH-g dishes, we anticipated that the reprogramming efficiency into hiPSC will be enhanced under W10% HSt-PMEDSAH-g culture conditions. As we have previously reported [18], here we used non-integrating Sendai virus constructs to overexpress OCT4, SOX2, KLF4, and C-MYC in human fibroblasts and reprogrammed them into hiPSC under a completely defined and xeno-free culture condition using W10% HSt-PMEDSAH-g dishes. Remarkably, we observed that the reprogramming efficiency was significantly increased when human somatic cells were reprogrammed into iPSC under the W10% HSt-PMEDSAH-g culture condition compared to the Matrigel control group.

## 5. Conclusions

We have established a chemically defined and xeno-free culture condition for the long-term maintenance and derivation of hPSC using our optimized and standardized 10% HSt-PMEDSAH-g dishes. The karyotype of hPSC remains stable after long-term culturing in vitro. Furthermore, the chemically defined and xeno-free culturing system also enables the efficient derivation of hiPSC from human fibroblasts through reprogramming. The culture conditions described here are ideal for reducing lot-to-lot variation and increasing reproducibility, which can ensure consistency for the use of hPSC in both basic research and clinical applications. The long-term maintenance of hPSC without compromising safety and functionality in this defined culture system will reduce the economic burden of their large-scale expansions for various applications. Our culture system has promising potential for the extensive applications of hPSC in in vitro studies to better understand developmental biology and pave the way for translating hPSC technology to the clinic.

## Figures and Tables

**Figure 1 bioengineering-10-00999-f001:**
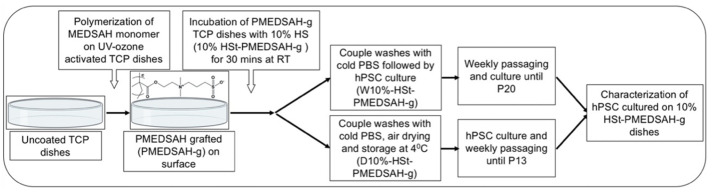
Schematic representation of the experimental protocol. PMEDSAH grafted (PMEDSAH-g) TCP dishes were synthesized using a surface-initiated graft polymerization procedure. The dishes were then pre-conditioned/treated with 10% human serum (HSt) (*v*/*v* DMEM/F12) and separated into two sets of 10% HSt-PMEDSAH-g dishes. The first set (W10% HSt-PMEDSAH-g) was used right after its preparation for culturing hPSC for 20 consecutive passages (wet-W). The other set (D10% HSt-PMEDSAH-g) was air dried and wrapped with parafilm, stored at 4 °C until their use for cell culture (dry-D), and used for the weekly passaging of hPSC for 13 consecutive passages. In both conditions, the cells were characterized for self-renewal and pluripotency.

**Figure 2 bioengineering-10-00999-f002:**
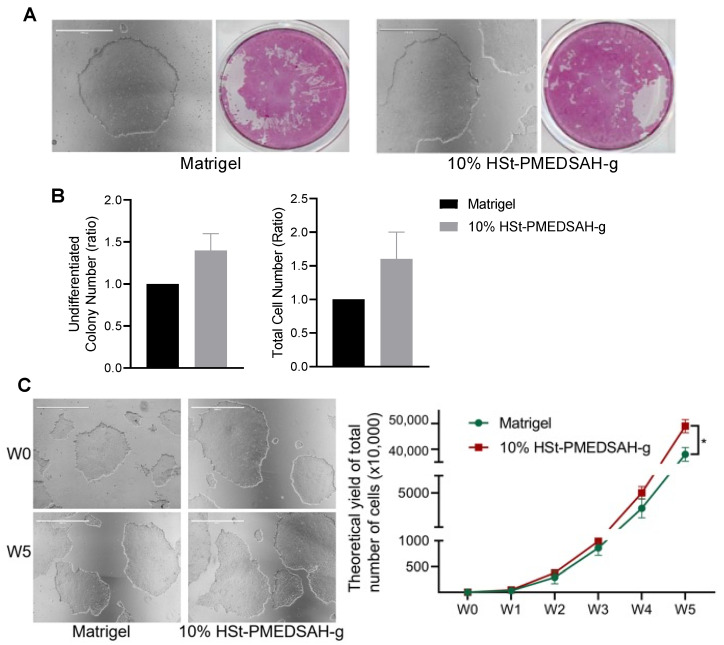
Culture and expansion of hPSC on W10% HSt-PMEDSAH-g dishes. Fresh PMEDSAH-g dishes treated with 10% HS (W10% HSt-PMEDSAH-g) in DMEM/F12 gave better results for promoting and stabilizing cell attachment and long-term culture of undifferentiated hPSC compared to cells cultured on Matrigel-coated dishes. (**A**) Representative micrographs of hPSC colonies cultured on Matrigel-coated plates and W10% HSt-PMEDSAH-g. On the right side, respective alkaline phosphatase (AP) stained colonies. (**B**) A plot of undifferentiated colony number (ratio) and total cell number (ratio) compared to the control group (Matrigel-coated dishes) indicating that W10% HSt-PMEDSAH-g lead to a higher number of undifferentiated colonies and the total number of cells on week 1. (**C**) Representative micrographs of hPSC colonies cultured on W10% HSt-PMEDSAH-g and Matrigel-coated plates during week 0 and week 5. The graph to the right indicates the theoretical yield of the total number of cells counted and compared from colonies cultured on W10% HSt-PMEDSAH-g and Matrigel-coated dishes during weekly passaging until passage 5. Scale bars, 1000 µm. * *p* < 0.05, (*n* = 3; unpaired *t* test). Error bars in graphs represent the SEM of the group.

**Figure 3 bioengineering-10-00999-f003:**
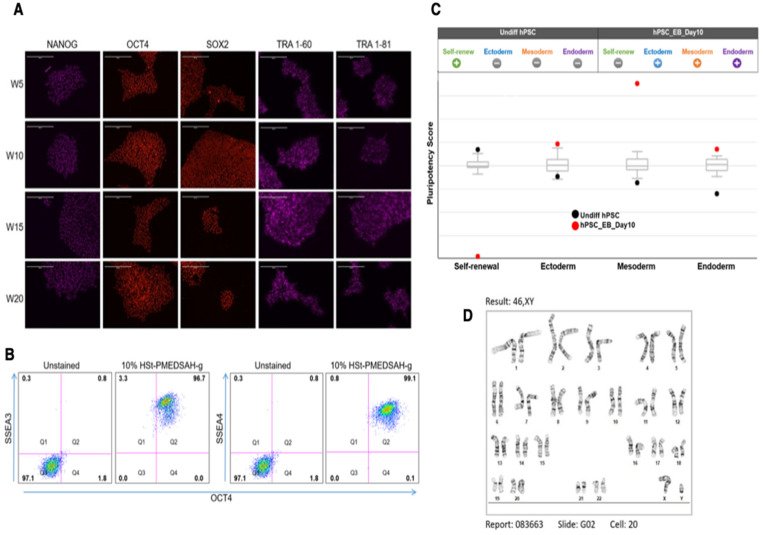
Characterization of hPSC after long-term culture on W10% HSt-PMEDSAH-g dishes. hPSC cultured on W10% HSt-PMEDSAH-g dishes for 20 consecutive passages, maintaining self-renewal and pluripotency. (**A**) Representative micrographs of immunocytochemistry every 5 weeks showing strong positive staining for human pluripotency markers NANOG, OCT4, SOX2, TRA1-60, and TRA-181. (**B**) Dot plots of hPSC after 20 passages of culture on W10% HSt-PMEDSAH-g dishes, showing more than 99% co-expression between SSEA3/OCT4 and SSEA4/OCT4. Isotype controls for the respective antibodies were used. The threshold/gate was set to a maximum of 0.8% positive cells in the unstained control and every signal above was counted as a positive signal. FMO controls were used to assess the spread of the fluorochrome into the missing channel and the gates were set accordingly. (**C**) Pluripotency was confirmed by qPCR hPSC ScoreCard assay quantifying the self-renewal and trilineage differentiation potential of passage 20 hPSC cultured on W10% HSt-PMEDSAH-g and 10 days old EB developed from the same pool of cells. (**D**) G-banding karyogram of cells cultured on W10% HSt-PMEDSAH-g dishes at passage 20 confirming genetic integrity. Scale bars, 200 µm.

**Figure 4 bioengineering-10-00999-f004:**
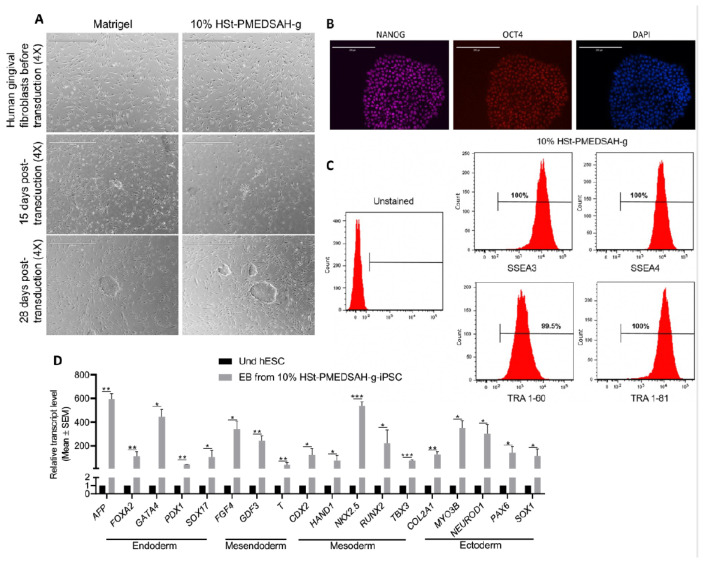
Reprogramming of human gingival fibroblasts into induced PSC (iPSC) on W10% HSt-PMEDSAH-g dishes. Human gingival fibroblasts were reprogrammed into iPSC on Matrigel-coated dishes and on W10% HSt-PMEDSAH-g dishes. (**A**) Representative transmitted light micrographs showing fibroblast morphology before and during the development of hiPSC colonies, Scale bars, 1000 µm. (**B**) Representative micrographs of hiPSC developed on W10% HSt-PMEDSAH-g plates after immunocytochemistry staining with pluripotency-associated markers. DAPI was used as nuclear marker. Scale bars, 200 µm. (**C**) Representative histograms of hiPSC developed on W10% HSt-PMEDSAH-g dishes showing the expression of cell surface pluripotency-associated markers. Isotype controls for the respective antibodies were used. The threshold/gate was set to a maximum of 0.8% positive cells in the unstained control and every signal above was counted as a positive signal. (**D**) qPCR analysis for the expression of markers corresponding to all germ lineages showing trilineage differentiation potentiality of 10 days old EBs obtained from hiPSC developed on W10% HSt-PMEDSAH-g dishes and compared to undifferentiated hPSC. * *p* < 0.05, ** *p* < 0.005, *** *p* < 0.0005 (*n* = 3; unpaired *t* test). Error bars in graphs represent the SEM of the group.

**Figure 5 bioengineering-10-00999-f005:**
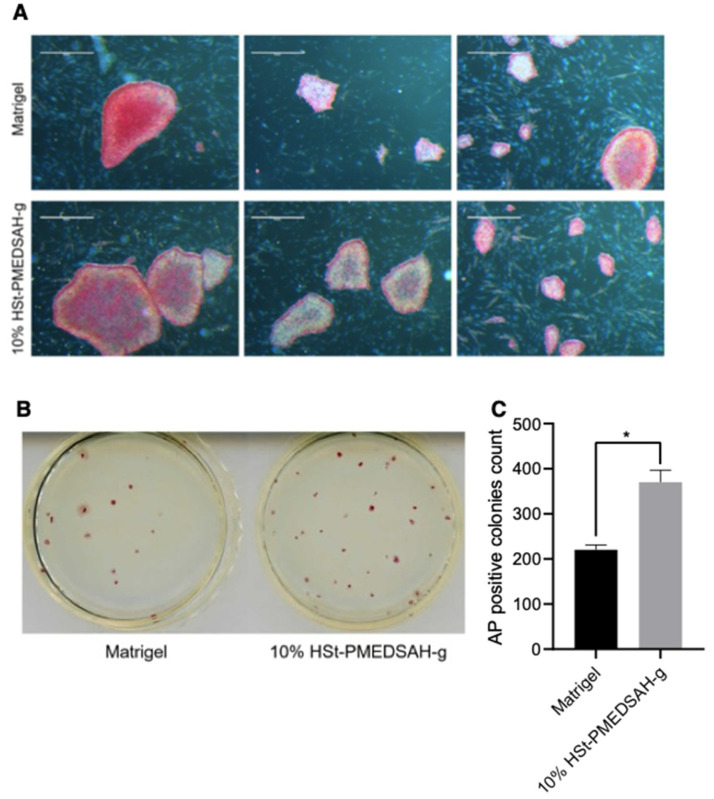
Improved reprogramming efficiency of human gingival fibroblasts into hiPSC on freshly prepared 10% HSt-PMEDSAH-g dishes.Human gingival fibroblasts were reprogrammed into iPSC on W10% HSt-PMEDSAH-g plates and Matrigel-coated plates. (**A**) Representative bright field images of iPSC colonies derived on W10% HSt-PMEDSAH-g and Matrigel-coated dishes after staining for alkaline phosphatase (AP). Scale bars, 5 mm. (**B**) Representative images of entire colonies in each condition and (**C**) graph indicating the average (N = 3) AP-positive colonies per culture dish in each condition. * *p* < 0.05, (*n* = 3; unpaired *t* test). Error bars in graphs represent the SEM of the group.

**Table 1 bioengineering-10-00999-t001:** List of supplements/chemicals systematically tested along with chemically defined commercial media to promote the attachment and long-term maintenance and growth of hPSC on PMEDSAH-g dishes. We analyzed the activation of integrins by using a peptide, integrin-activating antibody and by altering the pH of the chemically defined media. Supplementation of divalent cations and certain cytokines to the media were also analyzed. Analysis of treatment (pre-conditioning) of the PMEDSAH-g dishes with graded concentrations (1%, 10% and 100%) of fetal bovine serum (FBS), Knockout™ Serum Replacement (KOSR), and human serum (HS) were also included. For all these conditions, the observations on cell attachment and maintenance are shown. Interestingly, treating/pre-conditioning of the PMEDSAH-g dishes with 10% human serum (10% HSt-PMEDSAH-g) resulted in improved cell attachment and long-term culture of hPSC with chemically defined media.

Supplements Systematically Tested to Promote Cell Attachment				Concentration	Observations
Integrin Activation	Divalent Cations	Cytokines	Serum		
Commercial peptide (QHREDGS) [20,21]				10 nM	No cell attachment
Integrin activating antibody [22,23]				2 µg/mL	Low and unstable cell attachment on W0
Altering the pH of media used [24,25]				7.0, 7.2, 7.4, 7.6, 7.8, and 8.0	Low cell attachment, which was lost by W1
	Ca^2+^, Mg^2+^, Mn^2+^ [26,27]			3 mM	Cell attachment sustained until W3
		Phorbol-myristate acetate (PMA) [29]		100 ng/mL	No cell attachments
		Retinoic acid (RA) [30]		1 mM	Low and unstable cell attachment on W0
			Fetal Bovine serum (FBS) [31]	10%	Cell attachments and differentiation
			Knockout™ Serum Replacement (KOSR) [7]	10%	No cell attachments
			Human serum (HS) [3,33]	1%	Low cell attachment on W0, lost all cells by W4
			HS	10%	Consistently stable, and long-term cell attachment
			HS	100%	Highest cell attachment, but high degree of differentiation during each passage

## Data Availability

All relevant data are available with the article and its Appendix A, or available from the corresponding authors upon reasonable request.

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
