# Peer review of "Enhanced Expansion of Human Pluripotent Stem Cells and Somatic Cell Reprogramming Using Defined and Xeno-Free Culture Conditions"

_bioengineering, 2023, doi:10.3390/bioengineering10090999_

Round 1

Reviewer 1 Report

The authors describe a systematic study to identify the optimal conditions for long-term culture and maintenance of hPSC using the combination of synthetic substrates with chemically defined and xeno-free culture media. The article is relevant since ethical and scientific problems are present when using fetal calf serum.

The Article is well written and deserves publication with no  further revision

.

Author Response

Dear reviewer,

 Thank you for revising our manuscript and for appreciating our work aiming to develop improved culture conditions for hPSCs.

Reviewer 2 Report

Thanks for the authors’ submission titled "Enhanced expansion of human pluripotent stem cells and somatic cell reprogramming using defined and xeno-free culture conditions." I appreciate the effort the authors have put into investigating and improving the culture conditions for hPSC. Overall, the manuscript is well-structured and promising, but it would benefit from these clarifications in the context.

The abstract provides a good overview of the study and its relevance. However, there's a need for more detailed information regarding the methodology, results, and improvements over existing practices.

The phrases "a significant enhancement" and "better performance" are vague. For clarity and a stronger impact, consider quantifying the improvements and specifying what aspects of performance have been improved.

To increase the accessibility of the abstract to a broader audience, consider providing a brief explanation or the full form of specialized terms, such as "PMEDSAH".

The connection between the expansion of hPSC and the reprogramming efficiency of somatic cells into iPSC under the new culture conditions is not clear from the abstract. More explanation would help readers understand the full implications of the work.

Additional comment:

For Figure 5, please clarify the unit for the AP positive colonies count. Is it the number per area?

Author Response

Dear reviewer

Thank you for your kind comments and suggestions for our manuscripts. We appreciated that after incorporating them into our manuscript, our work has been improved. 

Please find specific responses accordingly in the abstract and in the legend of Fig 5. Those modifications can be identified in blue ink. 

Reviewer 3 Report

Major comments:

The manuscript by Timilsina et al focuses on an important topic for the fields of stem cell biology and regenerative medicine, which is the adoption of advanced xeno-free and chemically defined reagents for culture and maintenance of induced pluripotent stem cells iPSCs. Hinging upon their previous work on synthetic peptides as scaffolds for culturing those cells, here the authors explore the combination of synthetic peptides with human serum as an alternative path for culturing those cells while maintaining high quality metrics that would enable their path towards the clinic.

While the results appear to be positive, a major shortcoming of their strategy is that, due to its nature, human serum cannot be considered a chemically defined reagent. While this fact does not discredit their encouraging results, it should be highlighted more transparently, and explored in higher depth in the discussion. They should also offer insights on strategies to replace human serum with truly defined components.

Minor comments:

Fig 3A – Given that all antibodies possess a degree of specificity, unstained cells are not an acceptable negative control for the flow cytometry experiment. Ideally a biological negative control, such as a differentiated cell type (for example, commercially available hiPSC-derived cardiomyocytes) should be used. Alternatively, antibody isotypes can also be used as a technical rather than biological control, although they are less desirable than the proposed biological control.

Fig 4B – The images appear to be from independent cultures/areas. Ideally staining should be carried out in the same cell population, with the different markers overlayed against each other and with a nuclear marker such as DAPI.

Discussion

Authors accurately point out batch-to-batch variability of reagents as a source of complexity in PSC cultures. However their approach is based on the use of human serum, which will be inherently carry the same batch-to-batch variability. More detail should be given on how the authors hope to overcome this limitation.

Discussion, line 572: Due to its nature cannot be claimed to be a defined component.

Author Response

Dear reviewer, 

We appreciated your comments and suggestions to improve our manuscript.  We have addressed and incorporated all of them accordingly.  You can identify these modifications in the manuscript with  blue ink. 

Regarding the appropriate control for FACS analysis, now we provided the correct detail information about using the corresponding fluorochrome isotype as a technical control. We apologize for our initial mistake of not providing this information in the initial submission.  Following your suggestion, we also modified the micrographs in Fig. 4B, providing now, micrographs of the same colonies with different biomarkers and DAPI counterstain. You can find the new figure attached to this response.

Finally, we added a brief discussion about batch-to-batch inconsistencies that can be observed in human serum and our future work to overcome this limitation.  Thank you sincerely.
